# On the Colloidal Stability of Nitrogen-Rich Carbon Nanodots Aqueous Dispersions

**Thiago Fiuza [1,2], Guilherme Gomide [1], Alex Fabiano Cortez Campos [3], Fabrizio Messina [4] and Jérôme Depeyrot [1,\*]**

[1] Instituto de Física, Universidade de Brasília, 70904-970 Brasília, DF, Brazil; thiagofiuza91@gmail.com (T.F.); guilhermegomide@unb.br (G.G.)

[2] Laboratoire PHENIX, Sorbonne Université, UPMC Univ Paris 06, CNRS, Case 51, 4 Place Jussieu, F-75005 Paris, France

[3] Faculdade UnB Planaltina, Universidade de Brasília, 73345-010 Planaltina, DF, Brazil; relex@unb.br

[4] Dipartimento di Fisica e Chimica—Emilio Segrè, Università degli Studi di Palermo, Via Archirafi 36, 90123 Palermo, Italy; fabrizio.messina@unipa.it

\* Correspondence: jerome@unb.br; Tel.: +55-613-107-7818

**Abstract:** The present survey reports on the colloidal stability of aqueous dispersions of nitrogen-rich carbon nanodots (N-CDs). The N-CDs were synthesized by thermally induced decomposition of organic precursors and present an inner core constituted of a $\beta$-$C_3N_4$ crystalline structure surrounded by a surface shell containing a variety of polar functional groups. N-CDs size and structure were checked by combined analysis of XRD (X-ray Diffraction) and TEM (Transmission Electron Microscopy) measurements. FTIR (Fourier-Transform Infrared Spectroscopy) experiments revealed the presence of carboxyl and amide groups on N-CDs surface. Towards a better understanding of the relation between colloidal stability and surface charge development, zetametry experiments were applied in N-CDs dispersions at different pHs and constant ionic strength. The increase of the absolute values of zeta potential with the alkalinization of the dispersion medium is consistent with the deprotonation of carboxyl groups on N-CDs surface, which agrees with the macroscopic visual observations of long-term colloidal stability at pH 12. The saturation value of N-CDs surface charge density was evaluated by means of potentiometric-conductometric titrations. The difference between carboxyl-related surface charge and the one determined by zeta potential measurements point to the presence of oxidized nitrogen functionalities onto the N-CDs surface in addition to carboxyl groups. These novel results shed light on the electrostatic repulsion mechanism that allows for the remarkable colloidal stability of N-CDs dispersions.

**Keywords:** carbon nanodots; colloidal stability; surface charge

## 1. Introduction

Carbon nanomaterials have been under the spotlight of scientific development in numerous distinct areas over the last decades, notably carbon nanodots (CDs), considered 0-dimensional nanoparticles, offer exceptional bright [1] and tunable photoluminescence in the visible range [2]. Additionally, these low-cost and easily synthesized nanostructured materials [3] also incorporate appealing biophysical-chemical characteristics such as biocompatibility [4], low toxicity [5], good electron donor/acceptor [6] and high sensitivity to the environment [7]. The combination of these properties makes them suitable for the use in industry as nanosensors [8], photocatalytic and optoelectronic components [9,10], as well as on the medical field in bioimaging [11], photo-activated antiviral/antibacterial agents [12] and targeting nanotools applied to cancer therapy [13–15].

Generally, CDs are quasi-spherical nanoparticles smaller than 10 nm constituted of a carbonaceous core and a densely passivated disordered surface. Depending on the synthesis particularities, the core carbon atoms can exhibit different degrees of hybridization producing amorphous or crystalline structures [16]. The surface shell is also synthesis-dependent presenting a wide range of possible surface groups from small to very long atomic chains with hydrophilic or hydrophobic properties.

One of the most interesting properties of CDs is that they can be easily dispersed in water and other polar solvents up to large concentrations, leading to colloids with long-term stability [17]. This exceptional colloidal stability is found in as-synthesized CDs, without the need of any post synthesis functionalization, and is achieved thanks to the singular characteristic of the CDs surface, where functional groups provide an electric charge density that leads to effective electrostatic repulsions between particles, avoiding agglomeration phenomena [18]. The type of charge (positive or negative) imparted by the surface functional groups depends mainly on their chemical nature, on the pH and the ionic strength of the medium [19]. Groups with acid-base properties, such as amine and carboxyl, generate surface charge through protonation-deprotonation process, therefore the magnitude of the charge can be controlled by pH. Other groups provide constant structural charge, such as quaternary ammonium [20] (positive) and oxidized nitrogen functionalities [21] (negative). Possible specific adsorption of cations or anions also affects the surface charge and can even result in charge reversal [19]. Thus, at a constant temperature, the electrostatic repulsion between CDs can be tuned by varying the pH and the ionic strength of the dispersion, which corresponds to control the surface charge of the nanodots and the rate of screening in the electric double layer, respectively.

Among the large quantity of different approaches to obtain CDs, the popular microwave decomposition method has the advantage of enabling the nanoparticules doping with other atoms. In the context of this study, nitrogen atoms were introduced in the CDs carbon matrix originating the so called nitrogen-rich carbon nanodots (N-CDs). This special family has attracted great attention because of their big fluorescence quantum yields [5,22–24]. This bottom-up method also has the special feature of controlling the internal and surface structure of the CDs by balancing the proportion of the carbon and nitrogen molecular precursors. In sufficiently high N-doping conditions, the formation of a $\beta$-$C_3N_4$ crystalline core with a surface layer hosting amide, carboxyl and hydroxyl groups has been observed [4,25]. Thus, since amides and alcohols present $pK_a$ values in water usually higher than 15 [26], the charge of the N-CDs in aqueous colloidal dispersions arises mainly from deprotonation of surface carboxyl groups ($\equiv COOH_{(surf)}$) into carboxylate ($\equiv COO^-_{(surf)}$), in function of pH. The strength of acidity is related to the type of carboxyl group present at the N-CDs surface. pK values between 2.0 and 5.0 are typically associated with carboxylic acids [27], while pK values between 6 and 8 are attributed to hydrolyzed lactones or even carboxylic anhydride groups [28,29]. As a consequence of the heterogeneity of the carboxyl groups, the surface of N-CDs tends to exhibit pK values over a wide range, as commonly observed in carbon materials [30,31].

In previous studies [22,24,25,32], the structural and photoluminescent properties of N-CDs were extensively investigated, however very limited information is reported on the extent of the colloidal stability of these aqueous dispersions. The evaluation of the stability of the system presents major importance in order to appraise effects as local variations in nanoparticle concentration, change in interparticle interactions, aggregation and sedimentation processes. Understanding these phenomena should provide needful assistance, not only in the dispersibility, but also in the interpretation of the photoluminescence emission. As an example of the significance of this research, the origin of the CDs emission tunability has been a theme of intense debate in the literature [16], where aggregation effects are given a significant role on the data interpretation [33]. In this context, addressing the impact of physico-chemical parameters in surface properties would aid to clarifying interparticle interactions and ordering. Also, it could serve as an alternative or complementary procedure to the existing sorting methods (e.g. size exclusion chromatography [32] and ultracentrifugation [34]) to obtain more homogeneous characteristics making use of well controlled induced phase separations.

One of the key factors for stability in charged colloids is the development of the nanoparticles surface charge depending on the medium characteristics. Given the nature of the N-CDs charge mechanism in aqueous medium, we focused our attention on the pH-dependency, reasonably the most important parameter to monitor the surface charge. The limited range of studies on the stability of this material led us to develop a quantitative and qualitative analysis, by applying coupled experimental techniques to evaluate the surface charge density and link the results with the evolution over time of the macroscopical visual aspect of the samples. Also, a careful characterization of N-CDs in order to determine nanoparticles morphology, structure and sizes as well as to identify the nanoparticles main surface groups was required to guarantee a systematic and substantiated analysis.

The results reported on this study demonstrate the improvement of the colloidal stability in N-CDs dispersions as the pH of the aqueous medium undergoes a shift towards alkaline conditions. This response could be attributed to the increase in electrostatic repulsion based on the progression of the surface charge density. To fully comprehend the effect throughout the whole pH range, we propose a model combining potentiometric-conductometric titrations and zetametry results aiming the recognition of the different contributions to surface charge mechanism on the interface nanoparticle/solvent.

## 2. Materials and Methods

### 2.1. Reagents

The following pro analyse (P.A.) grade reagents, supplied from Sigma-Aldrich, Merck or Vetec Química Fina, were used for the nanoparticles synthesis and dispersion in aqueous media: citric acid monohydrate ($C_6H_8O_7 \cdot H_2O$), urea ($CH_4N_2O$), nitric acid ($HNO_3$ 63%), sodium hydroxide (NaOH) and sodium nitrate ($NaNO_3$). The aqueous solutions were prepared with deionized water Type I (Millipore Milli-Q Gradient quality).

### 2.2. Synthesis of the N-CDots Colloidal Dispersions

The synthesis of the nitrogen enriched carbon nanodots follows a thermally microwave-induced decomposition protocol performed in a household microwave oven. The procedure consists in the carbonization (microwave power set at 700 W) of an aqueous solution with completely dissolved citric acid monohydrate and urea. The organic precursor is prepared by diluting in 10 mL of Type 1 water 3 g of citric acid monohydrate with 3 g of urea, so that a 0.74 molar ratio of nitrogen and carbon (N/C) is established. After the complete water evaporation, the system is brought back to room temperature and can yield up to 30% in weight of the initial precursors. The reaction produces a strongly hygroscopic black powder consisting of aggregated N-CDots, which is dried under vacuum for 90 minutes ensuring the removal of any remaining solvent. Thenceforth, the powder can be dispersed in different aqueous solutions with well controlled physico-chemical parameters. The dispersion medium is primarily Type I water (pH ≈ 7), which had its pH regulated with either the addition of nitric acid or sodium hydroxide. The dispersion is performed by the addition of a certain mass of CDs to the corresponding volume of adjusted aqueous medium solution. The dispersion process is finalized with the use of an ultrasound bath at 50 °C until the macroscopic visual aspect of the sample is homogeneous.

### 2.3. Physical and Chemical Characterization

#### 2.3.1. Transmission Electron Microscopy

Transmission electron microscopy images were obtained in a JEM2100 electron microscope (JEOL) using a carbon type-A 400 mesh Cu substrate (TED PELLA). The N-CDs were dispersed in deionized water type I following the procedure described above. The dispersion was then diluted in alcohol and then deposed over the substrate and let to dry. The histogram was obtained by pixel measuring the

diameter of the N-CDs. A lognormal and gaussian function fit was performed in order to obtain the mean diameter of the size distribution.

### 2.3.2. X-ray Diffraction

X-ray diffraction was performed on post-synthesis dried powder, prior to any dispersion in aqueous media. The measurements were carried out in a D8 Focus (Bruker) diffractometer with Cu$K\alpha$ radiation (wavelength $\lambda$ = 1.5406 Å). The sample was probed in a 10°–90° range of $2\theta$ with a 0.05° step at 0.1°/min. A Si standard was used to obtain the instrumental line broadening $\beta_{ins}$. The nanoparticles mean size can be estimated by the broadening of the diffraction peaks using Scherrer's equation:

$$D_{XR} = \frac{k\lambda}{\beta \cos \theta},$$
(1)

where $k$ is the correlation factor (0.9 for spheres), $\lambda$ the x-ray beam wavelength, $\theta$ the incidence angle and

$$\beta = \sqrt{FWHM^2 - \beta_{ins}^2},$$
(2)

represents the line broadening coming from the dots characteristics determined with the full width at half maximum (*FWHM*).

### 2.3.3. Fourier-Transform Infrared Spectroscopy

Fourier transform infrared spectroscopy (FTIR) measurements were performed using a Perkin Elmer FTIR spectrometer (model Frontier) to gain further structural insights about the prepared N-CDs. Pellets of *KBr*-sample were prepared by mixing sample and potassium bromide then pressing at 10 tons on a hydraulic press (Auto-CrushIR, Pike Technologies). Spectra were recorded in absorbance mode from 4000 to 400 cm$^{-1}$ using 8 scans at 4 cm$^{-1}$ resolution.

### *2.4. Colloidal Stability of Dispersions*

### 2.4.1. Potentiometric and Conductimetric Titrations

The electrochemical measurements were acquired with a 713 Metrohm pHmeter with a pH glass double-junction electrode, a 712 Metrohm Conductimeter with a conductometer using a conductivity cell specially designed for colloidal dispersions and an electronic burette 665 Metrohm Dosimat. Potentiometric and conductimetric acid-base titrations were simultaneously performed on 35 mL of N-CDs dispersions at 0.15 g/L mass concentration in acidified (HNO$_3$) pH 2 aqueous medium. This mass fraction was selected in order to generate a well-resolved conductimetric curve where the distinction of strong and weak acids is evident. The titrations were performed from pH 2 to 12 by controlling the titrant (standardized NaOH solution, 0.01 mol/L, stirred and degassed by purified nitrogen during 10 minutes to avoid carbonation) addition with an electronic burette. Both potentiometer and conductometer were gauged using appropriate templates. Upon addition of the titrant, the pH and the conductivity of the sample medium are measured after equilibrium is reached. Thermal compensation is automatically applied in the measured conductivity. The titrations were performed in triplicate as usual in analytical chemistry and the results of equivalence points determinations present very low standard deviations. The titration analysis were performed by considering the deprotonation equilibrium of surface carboxyl groups according to:

$$\equiv COOH_{(surf)} + H_2O_{(l)} \overset{pK}{\rightleftharpoons} \equiv COO^-_{(surf)} + H_3O^+_{(aq)}$$
(3)

where $K$ is the corresponding thermodynamic constant ($pK = -\log K$).

### 2.4.2. Zetametry

Zeta potential measurements were taken using a ZetaSizer (model NanoZS 90, Malvern Panalytical, Malvern, UK) with a disposable folded capillary cell (DTS 1070). The N-CDs were dispersed in five different acidified ($HNO_3$) and alkalinized (NaOH) aqueous solutions: pH 2.0, 4.5, 7.0, 9.5 and 11.8. The ionic strength of 0.01 mol/L was imposed by adding $NaNO_3$ as a background electrolyte, avoiding the introduction of different ions to the system. The pH was measured before and after the addition of the N-CDs and in order to preserve the 0.15 g/L nanoparticles mass concentration, no adjustment of the physical-chemical parameters was performed after combining the N-CDs and the suited medium. Temperature variation/gradient effects were evaded by performing all measurements at 25 °C. The electrophoretic mobilities $\mu$ values are obtained from measurements of particle velocity using Laser Doppler Velocimetry (LDV). Then, the corresponding zeta potentials ($\zeta$) are calculated using the Henry's equation [35]

$$\mu = \frac{2}{3} \frac{\epsilon_0 \epsilon_r \zeta}{\eta} f(\kappa r) ,\tag{4}$$

where $\eta$ and $\varepsilon_r$ are the viscosity and dielectric constant of the solvent, respectively, and $\varepsilon_0$ is the vacuum permittivity. f($\kappa$r) is the Henry's function which depends on both Debye length $\kappa^{-1}$ and particle radius r. The Debye length is given by the expression

$$\kappa^{-1} = \left( \frac{\varepsilon_0 \varepsilon_r kT}{2 N_A e^2 I} \right)^{1/2} ,\tag{5}$$

where $I$ is the ionic strength of the dispersion. f($\kappa$r) varies from 1.0, for low values of $\kappa$r (Hückel model), to 1.5 as $\kappa$r approaches infinity (Smoluchowski model). For the transition range between low and high $\kappa$r, Ohshima [36] has provided an approximate analytical expression of f($\kappa$r) as

$$f(\kappa r) = 1 + \frac{1}{2} \left[ 1 + \frac{2.5}{\kappa r (1 + 2e^{-\kappa r)})} \right]^{-3} .\tag{6}$$

In addition, macroscopic visual observation of the evolution of the as prepared samples was carried out a week later. During this ageing process the samples were held motionless and at room temperature.

## 3. Results and Discussion

### 3.1. Physical and Chemical Characteristics of N-CDs

The characterization methods here employed allowed to confirm the expected morphology of the nanomaterial produced by referred synthesis protocol. TEM imaging is enable to resolve the N-CDs regardless the very small particle size, for instance, an aggregate of a few dots is exhibited in Figure 1 indicating the synthesis process successfully produced nanoparticles. Although the particles agglomerated due to the evaporation of the solvent, tracing boundaries is feasible owing to the internal structural organization of the N-CDs.

The crystalline structure is evidenced by the HRTEM measurements, where the lattice planes are visible and an almost spherical morphology is detected. TEM images are oftentimes not sufficient to determine the three-dimensionality of the particles, for example, it could be intricate to differentiate CDs from Graphene Quantum Dots (GQDs). However, the result here described is in strong agreement with high resolution transmission electron microscopy (HRTEM) combined with atomic force microscopy (AFM) reported in previous studies [22,24,32] with the same synthesis protocol, therefore endorsing the efficacy and reproducibility of the method. Due to their low contrast, it is a challenge to get enough images of N-CDs for size distribution histograms, but the observed N-CDs diameters lie mostly within the range of 2–4 nm.

Since TEM may have some limitations due to the nature of these nanoparticles, XRD was performed to obtain a mean size estimation by a different approach. The diffratogram presented

in Figure 2 shows one intense peak, with some lower intensity contributions that were subtracted since they were only slightly above noise level, and would not help from a qunatitative point of view. The absence of very sharp peaks assures that microcrystalline structures were not formed, an issue observed in higher N/C ratios Sciortino et al. [22].

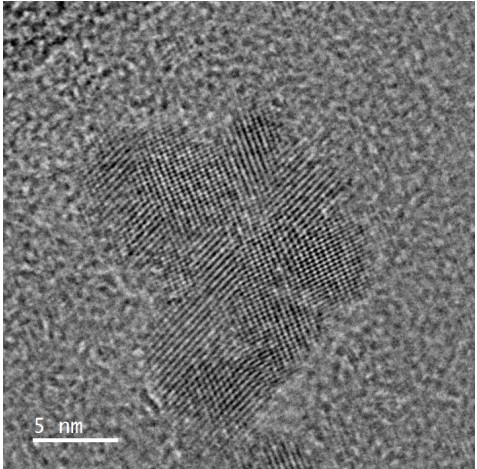

**Figure 1.** High resolution transmission electron microscopy of the studied sample, evidencing the crystalline structure of the dots.

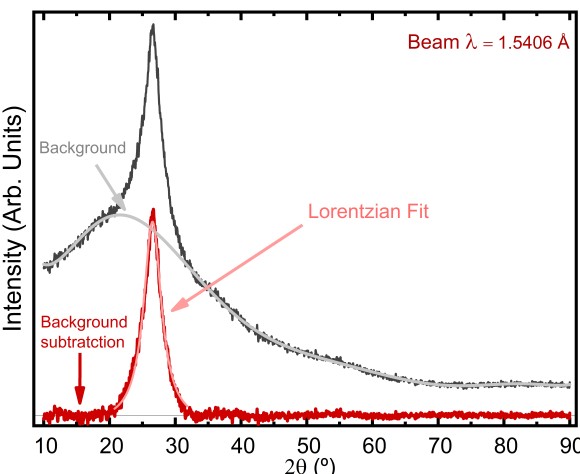

**Figure 2.** X-ray diffractogram of the investigated sample. Experimental values are shown in black and fitting results are shown in thinner solid lines.

The X-ray diffraction diameter ($D_{XR}$) is determined using Scherrer's equation and a particle diameter of 2.46 $\pm$ 0.03 nm is found. These results reinforce the already well established characterization of these nanoparticles as dots, namely, systems with really reduced particle size (1–8 nm) presenting a spherical symmetry. Thence, now the focus is directed over the surface features of this N-CDs, since the interface particle/solvent shall have a crucial role in the colloidal stability of these systems.

The mid-FTIR spectrum of the N-CDs is shown in Figure 3 and reveals their complex surface chemistry. The large broad band between 2600 and 3400 cm$^{-1}$ (A) are due to O–H, N–H and possibly C–H stretching vibrations. The peaks at 1712 cm$^{-1}$ (B) and 1600 $^{-1}$ (C) can be respectively attributed to carboxylic and amide C=O stretchings. The peak at 1401 cm$^{-1}$ (D) is characteristic of the O–H in plane bend while that at 1357 cm$^{-1}$ (E) can be assigned to the C–N stretching. Finally, the spectrum still shows characteristic H-O and H-N out of plane vibrations (F). These results are in agreement with the data presented on the article [24] upon the structural analysis of the N-CDs.

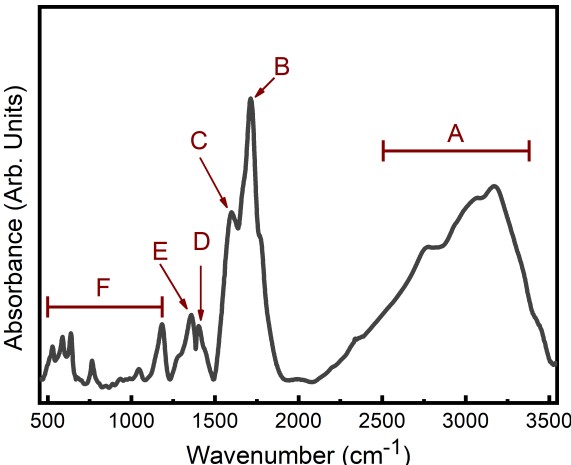

**Figure 3.** FTIR spectrum of the investigated sample, A-G areas illustrate the different regions of interest described in the text.

### 3.2. Colloidal Properties of N-CDs Aqueous Dispersions

Typical potentiometric-conductometric titration curves of N-CDs sample are presented in Figure 4, where the equivalence points $EP_1$ and $EP_2$ were determined by using direction lines applied to the conductometric curve [37–39]. These equivalence points delimit three distinct regions whose meaning can be described as follows. The first one corresponds to the strong acid titration (free $H_3O^+$ ions from $HNO_3$ in the bulk dispersion). The sharp decrease of the conductivity indicates the neutralization of $H_3O^+$ ions, gradually substituted by $Na^+$ ions from the titrant solution, which present lower specific molar conductivity [40]. After $EP_2$, the third region is related to the excess of titrant reagent, which increases the conductivity strongly due to the high specific molar conductivity of $OH^-$ ions. The second region, between $EP_1$ and $EP_2$, corresponds to the titration of N-CDs, i.e., the deprotonation of surface carboxyl groups. The contribution of N-CDs to the conductivity is negligible due to the mass of the particles, and the slight increase in conductivity can be more appropriately assigned to the increase of $Na^+$ ions concentration from the titrant.

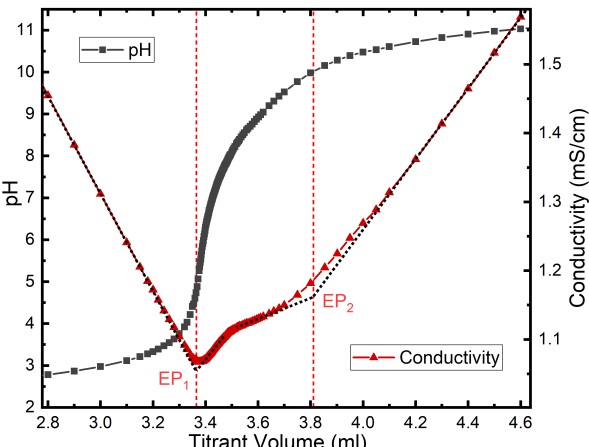

**Figure 4.** Potentiometric and Conductometric Titration showing the pH (in gray) and conductivity (in red) as a function of titrant volume. Vertical dashed lines indicate the equivalence points as determined from the conductivity curves.

As it can be observed, in this second region the potentiometric curve does not exhibit sharp changes, which is consistent with the presence of different surface carboxyl groups associated to various acidic constants [28,29,31]. Moreover, as previously mentioned, the protons of N–H and O–H bonds related amide and hydroxyl groups present at N-CDs surface are not titratable in the

investigated pH range [26]. Then, using the determined equivalence points and the mass balance, the total titratable acidity of the N-CDs surface could be calculated in terms of moles number of surface carboxyl groups per gram of N-CD sample ($[\equiv COOH_{(surf)}]_T$) leading to 9.8 mmol/g, a value similar to that of other carbon materials [41]. Considering the average diameter (2.46 nm) of our CDs, and the density of $\beta$-C$_3$N$_4$ (3.57 g/cm$^3$), one can calculate that this value of [COOH] corresponds to >100 carboxyl groups per dot, confirming the picture of CD surfaces very densely covered by such polar moieties.

The zeta potential of N-CDs dispersions obtained at different pH values are depicted in Figure 5 together with a picture that illustrates the macroscopic state of the colloidal systems after two weeks of preparation. As it can be seen, the zeta potential of N-CDs is strongly pH-sensitive. This behavior can be correlated with the deprotonation of carboxyl groups with increasing pH, which results in higher concentration of negatively charged surface groups and therefore more negative values of zeta potential. In charged colloids, as N-CDs, it is generally assumed that zeta potential higher than ǀ30 mVǀ indicates good stability against aggregation because of electrostatic repulsion between particles [42,43]. This is consistent with our experimental macroscopic observations of N-CDs dispersions at different pH values. Indeed, at pH = 11.8, the high negative zeta potential ensures a very stable sol and no phase separation is noted. For lower pH values, the zeta potential becomes less negative inducing coagulation and settling of nanoparticles at the bottom of the falcon tube. This aggregation effect is more pronounced at pH = 2.0, where a sharp precipitation of aggregated N-CDs can be observed.

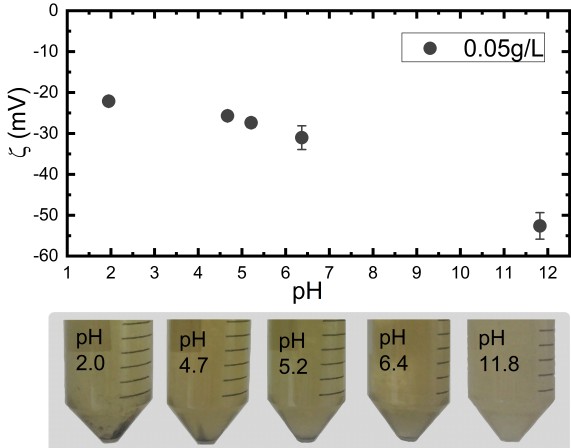

**Figure 5.** Top: Zeta potential as a function of the pH for the investigated N-CD sample in a concentration of 0.05 g/L. Bottom: Picture of the samples after two weeks in each pH, showing the reduction of precipitate amount with increasing pH.

Based on the titrations results, the pH-dependence of the concentration of acidic surface groups of N-CDs can be roughly evaluated in a model that considers that $[\equiv COOH_{(surf)}]_T$ is constant until EP$_1$. Then, the deprotonation process begins and the concentration of carboxylate surface groups $[(\equiv COO^-_{(surf)})]$ progressively increases until the pH corresponding to EP$_2$, where its maximum value is reached. Then, by applying the proton balance equation in the region between EP$_1$ and EP$_2$, the variation of $[\equiv COO^-_{(surf)}]$ with pH can be expressed as [44]:

$$\left[\equiv COO^-_{(surf)}\right] = \frac{1}{m}\left\{\left(V_0 + V_{EP_1}\right)\left(\left[H_3O^+\right]_{EP_1} - \left[OH^-\right]_{EP_1}\right) + V_t C_t \right.$$
$$\left. - \left(V_0 + V_{EP_1} + V_t\right)\left(\left[H_3O^+\right]_f - \left[OH^-\right]_f\right)\right\},$$

(7)

where m is the mass of N-CDs sample, $V_0$ is the initial volume of the N-CDs samples, $V_{EP_1}$ and $V_t$ are the volume of titrant added until $EP_1$ and from $EP_1$, respectively, $[H_3O^+]$ and $[OH^-]_{EP_1}$ are the molar concentration of $H_3O^+$ ions and $OH^-$ ions at $EP_1$, $C_t$ is the molar concentration of titrant and $[H_3O^+]_f$ ($[OH^-]_f$) are the equilibrium concentration of $H_3O^+$ ($OH^-$) ions. In this model, the surface charge density of carboxylate groups ($\sigma_{COO^-}$) can be estimated as

$$\sigma_{COO^-} = \frac{\left[\equiv COO^-_{(surf)}\right] m N_A e}{A_T}, \tag{8}$$

where $N_A$ is Avogadro's constant, $e$ the elementary charge and $A_T$ corresponds to the total surface area of nanoparticles, calculated considering spherical N-CDs of diameter $D_{XR}$. In order to cross the analysis of the potentiometric-conductometric titrations with that of zeta potential measurements, we have calculated the electrokinetic charge density ($\sigma_\zeta$) through the following empirical formula [45–47]:

$$\sigma_\zeta = \frac{\varepsilon_0 \varepsilon_r kT}{e} \kappa \left[2\sinh\left(\frac{e\zeta}{2kT}\right) + \frac{4}{\kappa r}\tanh\left(\frac{e\zeta}{4kT}\right)\right], \tag{9}$$

where $k$ is the Boltzmann constant and $T$ the temperature. On one hand, $\sigma_{COO^-}$ is directly proportional to the concentration of carboxylate groups onto N-CDs surface, therefore, in absence of specific adsorption, this parameter can be finely tuned by the pH. On the other hand, $\sigma_\zeta$ is particularly dependent on the ionic strength of the dispersion since the electrokinetic charge density is defined as the effective electric charge normalized on the area of the slip plane in the electric double layer [48].

Figure 6 shows the pH-dependence of $\sigma_{COO^-}$ and $\sigma_\zeta$ evaluated using Equations (8) and (9), respectively. As it can be seen, the profile of the curves obtained from the two measurements exhibits the same general trend, but the absolute value of the charge densities at a fixed pH shows a sharp difference. Based on the acidity of the particles surface, in the sample of N-CDs dispersed at pH = 2.0, the concentration of carboxylate groups is zero, leading to $\sigma_{COO^-} = 0$. However, the particles present a negative electrokinetic charge density at this pH ($\sigma_\zeta = -0.019$ C/m$^2$), which can be attributed to the presence of oxidized nitrogen functionalities onto N-CDs surface, as already observed for similar carbon materials [21,49,50]. As a matter of fact, XPS studies of these CDs [24] revealed the presence of N−O bonds on the surface, confirming the present conclusion. In the other samples, the N-CDs present increasing $|\sigma_{COO^-}|$ values with pH, due to the deprotonation of carboxyl groups, as previously discussed. Nevertheless, the increase of $|\sigma_\zeta|$ with pH is less significant because of the screening effect generated by the excess of counterions from the electrolytes (acid or base) used to adjust the pH of the medium [51].

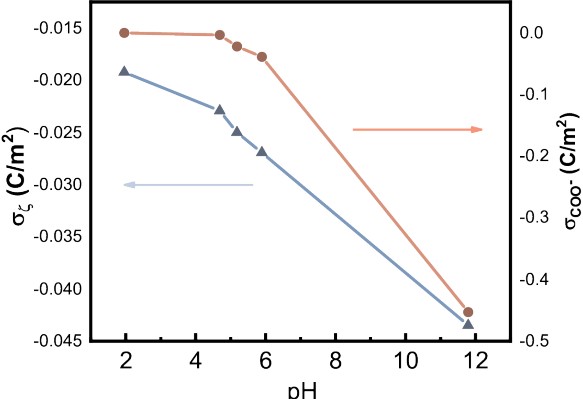

**Figure 6.** Electrokinetic charge density $\sigma_\zeta$ (blue triangles) and calculated surface charge density of carboxyl groups $\sigma_{COO^-}$ (red circles), both in C/m$^2$.

## 4. Conclusions

We have successfully synthesized nitrogen-rich carbon nanodots (N-CDs) and dispersed them in water in order to investigate the mechanisms that play a role in their colloidal stability. The presence of carboxyl and amide groups on N-CDs surface were revealed by FTIR experiments. A well characterized sample of N-CDs with a $\beta$-C$_3$N$_4$ core was used to maintain the focus on the surface charge mechanisms, which are responsible for controlling the aggregate formation and, therefore, affects the fluorescence tunability and efficiency.

Colloids zetametry in different pH was carried out to relate the colloidal stability to the development of surface charge due to the polar functional groups. The enhanced dispersibility of the N-CDs at high pH results from the high concentration of carboxylate groups on the surface that induces electrostatic repulsion between particles. The increase of the absolute value of the zeta potential with alkalinization is consistent with the deprotonation of these carboxyl groups on the N-CDs surface and explains the visual observations of improved colloidal stability at higher pH.

However, the carboxyl groups deprotonation alone cannot explain the negative zeta potential values we find at pH 2, since they do not contribute to the surface charge of N-CD at low pH. This negative electrokinetic charge density value determined at pH 2.0 can be attributed to the presence of oxidized nitrogen functionalities onto the N-CDs surface. The comparison of the electrokinetic charge density and the surface charge density of carboxyl groups allowed us to better understand the role of the surface groups on the pH-dependence of the zeta potential and, consequently, on the electrostatic repulsion mechanism that guarantees the colloidal stability of the investigated carbon nanodots dispersions.

**Author Contributions:** Conceptualization, T.F., G.G., A.F.C.C., F.M. and J.D.; methodology, T.F., G.G. and A.F.C.C.; validation, A.F.C.C., F.M and J.D.; formal analysis, T.F., G.G., A.F.C.C.; investigation, T.F., G.G. and A.F.C.C.; resources, A.F.C.C., F.M. and J.D.; writing—original draft preparation, T.F. and A.F.C.C.; writing—review and editing, G.G., F.M and J.D.; visualization, T.F. and G.G.; supervision, F.M. and J.D.; project administration, A.F.C.C. and J.D.; funding acquisition, A.F.C.C. and J.D.

**Funding:** Authors gratefully acknowledge the financial support of the Brazilian agencies: The Coordination for the Improvement of Higher Education Personnel (CAPES), the National Council for Scientific and Technological Development (CNPq – Grant Number 465259/2014-6 and 400849/2016-0), the National Institute of Science and Technology Complex Fluids (INCT-FCx) and the Distrito Federal Research Foundation (FAP-DF – Grant Number 0193.001569/2017, 0193.001194/2016 and 0193.001376/2016).

**Acknowledgments:** The authors would like to thank T. Oliveira dos Santos and Laboratório Multiusuário de Microscopia de Alta Resolução (LabMic) of University of Goiás, Brazil, for HRTEM measurements.

**Conflicts of Interest:** The authors declare no conflict of interest.

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
