# Peer review of "On the Colloidal Stability of Nitrogen-Rich Carbon Nanodots Aqueous Dispersions"

_carbon_

Round 1
Reviewer 1 Report
The paper presents an interesting and systematic characterization of N-doped Carbon dots and their properties in water dispersions. However the content of the paper is in my opinion too meager for a publication, owing to the previous similar publications reported by the same authors (i.e. Refs. 22, 24, 25, 34...).
In the introduction, the original content of the paper and its contribution to the field is lacking and must be in constrast clearly stated, also comparing to the previous literature cited above.
Furthermore, the paper reports the results of only one type of sample, although it would be interesting to show some parameter variations/comparisons among samples in terms of, for example, synthesis conditions, doping amount or type....
Some proof-of-concept validations or applications, i.e. luminescence or other data showing the significance of the research in an applied field, are needed.
Reviewer 2 Report
I think there is no need to write in experimental part where (in which laboratory) the measurements were done.
Page 2 line 35 : "is" is doubled
Page 2 line 52 : grat or big instead of large
Page 3 lines 78-81 : It would be good to give concentrations of urea and citric acid in solution subjected to carbonization not only the rato N:C.
Page 4 lines 1106-109 : It would be good to show equations (Scherrer's and another one) in separated lines with numbers as it is for further ones. What is FWHM ?
Page 5 line 134 : Equation for Debye length should be placed in separate line with number.
Page 9 and 10 : In the text it is written that samples are visualized after 48 hours (so 2 days) after preparation on Fig. 5 and in the description under Fig. 5 the time is 2 weeks. What is true ?
Page 9 line 211 : It is not fluid-solid phase transition but just precipitation af aggregated CDs.
Round 2
Reviewer 1 Report
The authors have addressed my concerns from prior review.